



# Validation of the Cloud_CCI cloud products in the Arctic

Kameswara S. Vinjamuri[1], Marco Vountas[1], Luca Lelli[1,2], Martin Stengel[3], Matthew D. Shupe[4,5], Kerstin Ebell[6], and John P. Burrows[1]

[1]Institute of Environmental Physics, University of Bremen, Bremen, Germany
[2]Remote Sensing Technology Institute, German Aerospace Centre (DLR), Wessling, Germany
[3]Deutscher Wetterdienst, Offenbach, Germany
[4]Cooperative Institute for Research in Environmental Science, University of Colorado, Boulder, USA
[5]Physical Sciences Laboratory, National Oceanic and Atmospheric Administration, Boulder, USA
[6]Institute for Geophysics and Meteorology, University of Cologne, Cologne, Germany

**Correspondence:** K. S. Vinjamuri (kamesh@iup.physik.uni-bremen.de)

**Abstract.** The role of clouds in the Arctic radiation budget is not well understood. Ground-based and airborne measurements provide valuable data to test and improve our understanding. However, the ground-based measurements are intrinsically sparse, and the airborne observations are snapshots in time and space. Passive remote sensing measurements from satellite sensors offer high spatial coverage and an evolving time series, having lengths potentially of decades. However, detecting clouds by passive

satellite remote sensing sensors is challenging over the Arctic because of the brightness of snow and ice in the ultraviolet and visible spectral regions, and because of the small brightness temperature contrast to the surface. Consequently, the quality of the resulting cloud data products needs to be assessed quantitatively. In this study, we validate the cloud data products retrieved from the Advanced Very High Resolution Radiometer (AVHRR) post meridiem (PM) data from the polar-orbiting NOAA-19 satellite and compare them with those derived from the ground-based instruments during the sunlit months. The

AVHRR cloud data products by the European Space Agency's (ESA) Cloud Climate Change Initiative (Cloud_CCI) project uses the observations in the visible and IR bands to determine cloud properties. The ground-based measurements from four high-latitude sites have been selected for this investigation: Hyytiälä (61.84° N, 24.29° E), North Slope of Alaska (NSA, 71.32° N, 156.61° W), Ny-Ålesund (Ny-Å, 78.92° N, 11.93° E), and Summit (72.59° N, 38.42° W). The Liquid Water Path (LWP) ground-based data are retrieved from microwave radiometers, while the Cloud Top Height (CTH) has been determined from

the integrated lidar-radar measurements. The quality of the satellite products, Cloud Mask and Cloud Optical Depth (COD), have been assessed using data from NSA, whereas LWP and CTH have been investigated over Hyytiälä, NSA, Ny-Å, and Summit.

The Cloud_CCI COD results for liquid water clouds are in better agreement with the NSA radiometer data than those for ice clouds. For liquid water clouds, the Cloud_CCI COD is underestimated roughly by 2.8 Optical Depth (OD) units. When ice

clouds are included, the underestimation increases to about 4.6 OD units. The Cloud_CCI LWP is overestimated over Hyytiälä by $\approx$ 7 g$\,$m$^{-2}$, over NSA by $\approx$ 16 g$\,$m$^{-2}$, and over Ny-Å by $\approx$ 24 g$\,$m$^{-2}$. Over Summit, CCI LWP is overestimated for values $\leq$ 20 g$\,$m$^{-2}$ and underestimated for values > 20 g$\,$m$^{-2}$. Overall the results of the CCI LWP retrievals are within the ground-based instrument uncertainties. For CTH retrievals, the Cloud_CCI product overestimates the CTH for single-layer clouds. To understand the effects of multi-layer clouds on the CTH retrievals, the statistics are compared between the single-layer clouds





and all types (single + multi layer). When the multilayer clouds are included (i.e., all types), the observed CTH overestimation become underestimations of about 360-420 m. The CTH results over Summit station showed the highest biases compared to the other three sites. To understand the scale-dependent differences between the satellite and ground-based data, the Bland-Altman method is applied. This method does not identify any scale-dependent differences for all the selected cloud parameters except for the retrievals over the Summit station. In summary, the Cloud_CCI cloud data products investigated agree reasonably well

with those retrieved from ground-based measurements, made at the four high-latitude sites.

## 1 Introduction

Over the past decades, knowledge of cloud properties in the melting Arctic limited our understanding of their role in climate change (Devasthale et al., 2020). Clouds influence the earth's radiation budget by scattering the incoming solar radiation and absorbing and emitting outgoing longwave radiation in the thermal infrared. Clouds at different heights and temperatures

also have different effects on the incoming shortwave (SW) and outgoing longwave (LW) radiation. The net amount of solar radiation reaching the earth's surface depends on the optical properties of clouds and their horizontal and vertical geometrical extent. For example, the low-level cloud fraction from 2000 to 2016 has increased over the sea ice regions of Beaufort and East Siberian Seas during the summertime and resulted in an increasing ice melt. This behavior strengthens the ice-albedo feedback in the Arctic (Huang et al., 2021). At least since the beginning of the 1990s, Arctic surface temperature has been increasing

more rapidly than at lower latitudes, a phenomenon called Arctic Amplification, AA (Wendisch et al. (2017) and references therein). Long-term data records of cloud properties are required to understand the Arctic clouds and their impacts on AA's causes, and consequences. Cloud properties, measured at the ground-based sites, which are sparse, can be complemented by satellite retrievals to improve the coverage of observations. Improved knowledge of the spatial and temporal variation of clouds micro-physical (e.g., Cloud Optical Depth (COD), Liquid Water Path (LWP)) and macro-physical properties (e.g., Cloud

Fraction (CF), Cloud Top height (CTH)) in the Arctic improve our understanding of cloud processes and radiation balance (e.g., He et al. (2019); Devasthale et al. (2020); Lelli et al. (2022)). Cloud data products from satellite remote sensing provide daily global coverage and high spatial resolution over the Arctic. In addition, these cloud data products can help to evaluate climate models and assess anomalies in the reanalysis data products (Liu and Key, 2016; Shaw et al., 2021). Thus the accuracy of these cloud data products needs to be assessed and validated in the Arctic. However, retrievals of cloud properties from measurements

by passive remote sensing instruments over snow and ice is challenging and thus error-prone. These errors are mainly because of the low spectral contrast between the cloud and the surface in the visible spectral region and temperature inversions which influence the radiance observed in the near-infrared and thermal infrared spectral regions (see Dybbroe et al. (2005), Marchand (2016) and references therein). In addition, multilayer clouds occur in the Arctic, leading to issues in interpreting the cloud parameters (e.g. CTH), resulting in biases. The lack of validation studies for the satellite cloud data products at high latitudes

may have limited their use by the scientific community undertaking research in the Arctic. There are some exceptions, such as the use of cloud cover (Schweiger, 2004; Boccolari and Parmiggiani, 2018; Philipp et al., 2020), which have investigated cloud trends and variability over the Arctic. In addition, some studies, such as Sporre et al. (2016), have compared LWP and



CTH retrievals from Moderate Resolution Imaging Spectroradiometer (MODIS) and Visible Infrared Imaging Radiometer Suite (VIIRS) with the measurements of the Atmospheric Radiation Measurement (ARM) mobile facility at the high-latitude site in Hyytiälä. They found reasonable agreement between LWP and CTH data products: one exception being CTH from thin cirrus clouds. Earlier versions of the MODIS measurements were compared with ARM measurements over the North Slope of Alaska (NSA), where the authors identified cloud detection problems at high viewing angles (Berendes et al., 2004). Another study by Liu et al. (2017) examined active satellite observations in the Arctic compared to two ground-based stations, NSA and Eureka, and found from the cloud fractions that the satellite observations show 25-40% fewer near surface clouds (< 0.5 km).

In this study, we validate the satellite cloud retrieval using selected cloud parameters, i.e., cloud mask, COD, LWP, and CTH, from the Cloud Climate Change Initiative (Cloud_CCI) dataset with ground-based cloud data products. The Cloud_CCI is a recent global data set of cloud properties with good coverage in the Arctic. The Cloud_CCI data products are retrievals from the measurements made by the Advanced Very High Resolution Radiometer (AVHRR) on a series of satellite platforms (Stengel et al., 2020). It is part of the European Space Agency's program to provide climate data records within the United Nations Convention on Climate Change framework. In this study, we use the data products retrieved from the AVHRR on NOAA-19 daytime measurements made during the sunlit months, specifically April, May, June, July, August, and September for the time-period 2010-2019. NOAA-19 flies in a sun-synchronous orbit in an ascending node, having an equator crossing time of 13:30 hrs. Previously, validation of the Cloud_CCI data set has typically focused on latitudes outside of the Arctic (see Jeanneret et al. (2018)). In this study, ground-based measurements from one sub-Arctic site at Hyytiälä Finland (61.84° N, 24.29° E) and three Arctic sites at NSA (North Slope of Alaska) USA (71.32° N, 156.61° W), Ny-Ålesund Svalbard (78.92° N, 11.92° E), and Summit Greenland (72.59° N, 38.42° W) are used to validate the Cloud_CCI data product. All these sites have similar ground instrumentation. The measurements made at Hyytiälä were part of the Biogenic Aerosols – Effects on Clouds and Climate (BAECC) campaign and are available for 2014. In contrast, those at NSA and Summit provide a relatively long-term data record from 2010-2018 that matches the data record from the NOAA-19 AVHRR. The ground-based data record from Ny-Ålesund (Ny-Å) used in this study covers the years 2016-2018. The Hyytiälä and NSA ground-based data are from the ARM research facilities. The ARM program, initiated by the U.S. Department of Energy, provides atmospheric measurements from targeted campaigns and also long-term measurements at some sites. ARM cloud data products have been used to validate satellite cloud retrievals (Rutan et al., 2001; Ji and Shi, 2012; Sporre et al., 2016). The data from Summit were obtained from the Integrated Characterization of Energy, Clouds, Atmospheric state, and Precipitation at Summit, ICECAPS, a project funded by the U. S. National Science Foundation. For Ny-Å, measurements are taken from the AWIPEV atmospheric observatory. While some measurements at AWIPEV, e.g. radiosondes, surface radiation and meteorology, have started more than 30 years ago, cloud observations have been enhanced in 2016 as part of the Transregional Collaborative Research Center TRR172 "ArctiC Amplification: Climate Relevant Atmospheric and SurfaCe Processes, and Feedback Mechanisms (AC)[3]" (Wendisch et al., 2022). In particular with the installation of a cloud radar, enhanced vertically resolved cloud observations became possible (Nomokonova et al., 2019).



The satellite retrievals are often subject to a scale bias (i.e., scale-dependent bias) compared to the measurements from ground-based instruments. They tend to overestimate (underestimate) the high (low) values of COD, LWP, and CTH (Fischer et al., 2000; Sporre et al., 2016). Consequently, in the present study, we investigated the scale bias between the selected Cloud_CCI data products and those from the ground-based measurements in the Arctic. For this purpose, we use the Bland-Altman (BA) approach (Altman and Bland, 1983; Bland and Altman, 1986; Knobelspiesse et al., 2019). The BA approach uses the normalized bias for the paired mean values between the ground-based measurements and the Cloud_CCI data for the selected cloud parameter.

In section 2 of the paper, we introduce the Cloud_CCI cloud data products and then the corresponding data product from the ground-based observational sites. Section 3 describes the method used to assess and validate the Cloud_CCI data products, i.e., the statistical techniques. The results and their discussion are presented in section 4. In the last section, we summarize the findings of this study.

## 2 Data Sources

### 2.1 CCI data

As described above, we use the Cloud_CCI dataset retrieved from the measurements of AVHRR which flew on NOAA-19. The interested reader is referred to the detailed publication about the data by Stengel et al. (2020). The Cloud_CCI retrievals use the AVHRR radiance measurements centered at the wavelengths 0.6, 0.8, 3.7, 11, and 12 $\mu$m. The version 3.0 Level 3U daily products used in this study contain selected Level-2 data (satellite pixel level data) on a global 0.05° longitude-latitude grid (Stengel et al., 2019, 2020). The Cloud_CCI products are retrieved using the Community Cloud retrieval for CLimate (CC4CL) retrieval system (Sus et al. (2018); McGarragh et al. (2018)). It comprises three major steps involving:

– A cloud mask that identifies the cloudy and clear scenes is determined using an Artificial Neural Network (ANN). The ANN uses the AVHRR radiances trained with the co-located CALIOP COD to identify the presence or absence of cloud. In addition, a detection algorithm for cirrus clouds is used based on the brightness temperature differences in the IR band radiances at 11 and 12 $\mu$m and defined threshold values as proposed by Pavolonis et al. (2005).

– A cloud phase mask is used to determine liquid water and ice clouds using an ANN approach. The ANN cloud phase mask is similar to the cloud mask ANN, but it is trained with the CALIOP cloud top phase data. The cloud phase retrieval provides binary information, liquid or ice.

– An optimal estimation algorithm (Rodgers, 1976) is applied to the measured AVHRR radiances using the retrieved cloud mask and cloud phase information to generate the cloud data products. The ancillary information, such as land bidirectional reflectance distribution function (BRDF), surface temperature, and surface emissivity is taken from the MODIS C6 (Schaaf et al., 2010), ERA-Interim (Dee et al., 2011) and Cooperative Institute for Meteorological Satellite Studies (CIMSS) databases respectively (Seemann et al., 2008). Cloud properties retrieved using the optimal estimation

algorithm are cloud-top pressure, cloud effective radius (CER), and COD. These are used to calculate cloud liquid and ice water paths (Stephens, 1978). The COD and LWP are daytime products. One of the key features of the cloud CCI

dataset is the uncertainties provided at the pixel level inferred from the implemented optimal estimation theory (Stengel et al., 2020). The Cloud_CCI data has been validated by comparing it to independent space-borne references, especially for non-polar regions (Stengel et al., 2020; Stapelberg et al., 2019). The main focus of these studies was global-scale evaluation against space-based reference data as well as global-scale comparisons to existing datasets of similar types. The results presented in the article at hand extend the evaluation efforts by using high-quality, ground-based reference

measurements in the Arctic.

## 2.2 Ground based measurements

This study uses ground-based data from Hyytiälä, NSA, Ny-Å, and Summit stations shown in Fig. 1. Although of limited duration, the campaign data products, measured as part of the BAECC campaign (2014) over Hyytiälä provide relatively dense records with similar instrumentation and are directly comparable to measurements at the NSA (Verlinde et al., 2016), Ny-Å,

(Nomokonova et al., 2019) and Summit (Shupe et al., 2013) sites. Due to the availability of data products from the ground-based measurements, the CCI cloud mask (CM) and COD are validated at NSA, whereas CCI LWP and CTH are validated at all four sites.

### 2.2.1 Cloud fraction and optical depth

The cloud fraction (CF) and cloud optical depth over NSA are obtained from the radiative flux analysis product. This value-

added product is based on the shortwave irradiance from the pyranometer to identify clear and cloudy skies (Riihimaki et al., 2019). In the presence of clear skies, the shortwave irradiance, measured at the ground, is larger than that for cloudy skies. An empirical fit function is then used to calculate the continuous clear sky estimates and cloud optical property data products (Long and Ackerman, 2000; Recipes, 2007; Riihimaki et al., 2019). The CF is derived using the methodology described in Long et al. (2006), which assumes that increasing cloud cover increases the diffuse irradiance relative to the direct irradiance

(Riihimaki et al., 2019). The SW COD is retrieved using the algorithm from Barnard and Long (2004), which has been adapted for the identification of optically thin clouds (Barnard et al., 2008). The algorithm enables COD for liquid and ice clouds to be retrieved. The algorithm differentiates the liquid and ice clouds based on the fixed asymmetry parameter of 0.87 and 0.80, respectively, as suggested by Fu (1996); Barnard et al. (2008). The retrieved SW COD is only valid for overcast conditions, i.e., cloud fractions greater than 0.95 (Min and Harrison, 1996). The absolute difference in sky cover amount derived from the

SW CF estimates and total sky imager (TSI) retrievals agree within 10% indicating a good level of agreement (Long et al., 2006; Barnard et al., 2008).

### 2.2.2 Liquid water path

The LWP at Hyytiälä is retrieved from the 3-Channel Microwave Radiometer instrument (MWR3C) data product. The center frequencies of the microwave radiometer (MWR) are 23.83, 30, and 89 GHz, which coincides with the peak absorption fre-





quencies for the liquid water (Cadeddu, 2021). The use of the three channels makes the retrieval sensitive to the small LWP
values expected in dry environments, such as larger parts of the Arctic (Cadeddu et al., 2009; Cadeddu, 2021). The retrieval
uses an ANN approach as described by Cadeddu et al. (2009). The LWP at NSA is retrieved from an MWR measured bright-
ness temperatures at 23.80 and 31.40 GHz frequencies, while at Summit the same technique is applied to the same channels in
addition to a 90.0 GHz channel. For these two sites, a physical-based radiative transfer model for monochromatic light called

MonoRTM uses the observed measurements to create the MWR RETrieval (MWRRET) product (Turner et al., 2007; Gaustad
et al., 2011). The MWRs make measurements every 28 seconds and the random uncertainties associated with the retrieval for
MWR3C and MWRRET are $\approx$ 15 gm$^{-2}$ and $\approx$ 20 gm$^{-2}$ respectively (Turner et al., 2007; Cadeddu, 2021). For Ny-Å, LWP
is retrieved from the Humidity And Temperature PROfiler (HATPRO) of the Alfred Wegener Institute for Polar and Marine
Research (Nomokonova et al., 2019; Nomokonova et al., 2019). HATPRO is an MWR providing brightness temperature mea-

surements at 7 frequencies between 22.24 and 31.40 GHz and at 7 frequencies between 51.26 and 58.00 GHz with a temporal
resolution of about 1 s. LWP is retrieved from a linear regression algorithm (Nomokonova et al., 2019). The uncertainty is
estimated to be about 20-25 gm$^{-2}$ (Rose et al., 2005). In the analysis, HATPRO LWP has been excluded in cases where the
quality flag was set, i.e. indicating for example precipitation, and in cases where an additional check indicated that the MWR
brightness temperatures of one frequency channel were spectrally inconsistent with the other channels of the same band.

### 2.2.3 Cloud Top Height

CTH at Hyytiälä and NSA is measured using the Ka-band ARM Zenith-pointing cloud Radar (KAZR) Active Remote Sensing
of Clouds (KAZR-ARSCL) data product, which blends the radar observations with micropulse lidar observations (Clothiaux
et al., 2001) to provide, among other parameters, a detailed vertical cloud mask with cloud boundaries. The lidar provides
supporting information on the cloud boundaries, although for optically thick clouds the radar typically provides information

on the cloud top due to attenuation of the lidar signal. The CTH over Summit is derived from an earlier version of the ARSCL
algorithm but uses only observations from a Millimeter Cloud Radar (MMCR), which was the precursor for the KAZR and is an
operationally similar radar. Both radars used here are 35-GHz, Ka-band radars, which are commonly used for cloud detection
because the oxygen and water vapor absorption at these frequencies is at a local minimum (Clothiaux et al., 2001; Kollias et al.,
2007). The reflectivity in the range of -50 to 20 dBZ for the hydrometeor layer up to 10 km or higher is accurately detected

by the cloud radar (Clothiaux et al., 2000). The accuracy of the measurement for a stratocumulus or altocumulus cloud with
liquid particle size in the range of 3-5 $\mu$m at -50 dBZ is such that the detection limit for a hydrometeor layer is smaller than a
condensed water content of $\leq$ 0.01 gm$^{-3}$ and hydrometer layers in the range 0.01-0.03 gm$^{-3}$ are measurable (Noonkester, 1984;
Heymsfield et al., 1991; Clothiaux et al., 2001). The radar products have a vertical resolution of 45 m and a time resolution of 2-
4 sec. For Ny-Å, the Cloudnet classification product (Illingworth et al., 2007; Nomokonova et al., 2019) was used to determine

CTH. To this end, measurements at the AWIPEV atmospheric observatory, i.e. cloud radar reflectivity, Doppler velocity, and
ceilometer attenuated backscatter, are jointly analyzed with numerical weather prediction data. The resulting classification
profiles have a temporal (vertical) resolution of 30 s (20 m) and provide information on the presence of cloud liquid droplets,
ice, melting ice, and drizzle/rain in each radar height bin up to a height of about 12 km.



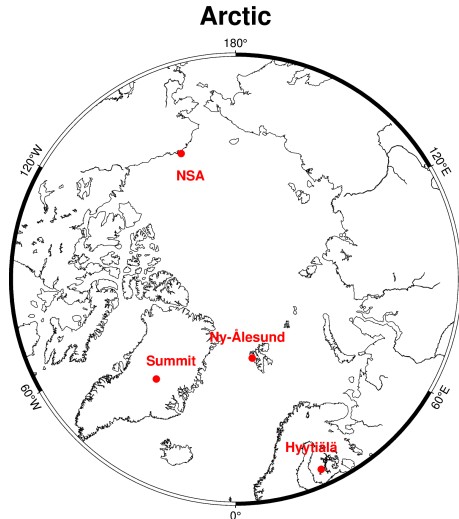

**Figure 1.** Arctic polar stereographic map showing the selected ground-based sites.

## 3    Method

One of this study's essential aspects is to match the satellite and the ground-based instruments spatio-temporally. We follow the spatial colocation strategy from Stengel et al. (2020) i. e., 5 km radius. However, in comparison to Stengel et al. (2020), we relaxed the temporal colocation from 3 to 5 minutes and averaged the ground-based data over this 5 minutes interval. We expect the satellite and ground-based instruments view the same cloud even at this relaxed interval, bearing in mind that the Arctic atmosphere is less dynamic than at the lower latitudes (Kay et al., 2016; Edel et al., 2019). For the quality criteria, the pixels

for the selected Cloud_CCI cloud data are not qualified for validation if the cost values are too high in the optimal estimation retrieval. The exact limits for each Cloud_CCI cloud parameter for the iteration process in the optimal estimation can be found in McGarragh et al. (2018). The validation uses the scatter plot and linear least-square regression coefficients to interpret the results. The reliability of the regression coefficients is examined using the t-distribution (Phillips, 1986). In the following text, the terms *overestimation* and *underestimation* refer to the median bias.

### 3.1    Bland-Altman Plots

Validation or assessment studies often use scatter plots to infer the difference or bias between the space and ground measurements. This method has its limitations (Martonchik et al., 2004; Van Harten et al., 2018; Seegers et al., 2018). For example, linear regression models do not provide information on scale-dependent biases. In addition to using scatter plots, we use Bland-Altman (BA) plots to further investigate the behavior of the bias. The BA approach used here plots the differences of the paired

measurements, normalized by the root mean square of the standard deviation sum versus the measurements' paired mean. This plot enables scale-dependent and scale-independent biases to be identified (Altman and Bland, 1983; Bland and Altman, 1986).





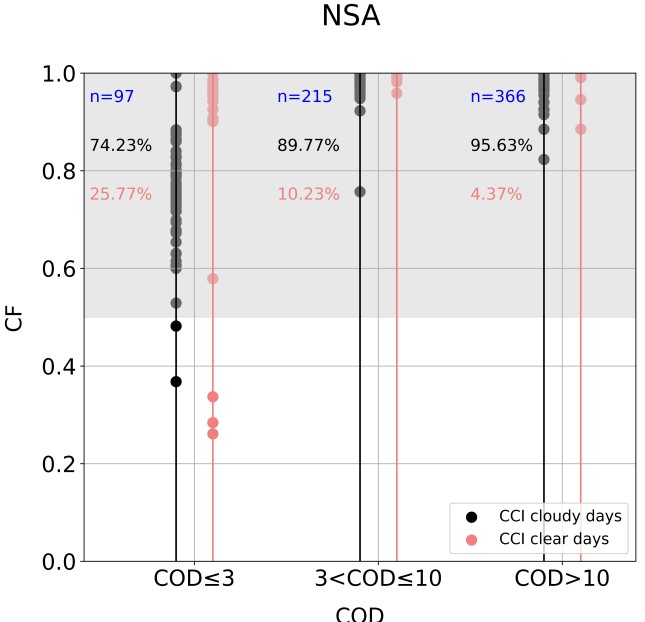

**Figure 2.** NSA cloud fraction (CF) vs. NSA cloud optical depth (COD) for different COD ranges for cloudy (black) and clear sky (red) stratified by Cloud_CCI measurements at NSA during the daytime in the period 2010-2018. The shaded grey region is the acceptance range, i.e., the range used to analyze the Cloud_CCI cloud mask accuracy. The value of 'n' represents the number of samples in the shaded region for that particular COD range. The percentage values written on the plot correspond to the hit (black) and miss percentages (red). More information is found in the text.

There have been a few relevant studies that have used this approach effectively (Knobelspiesse et al., 2019; McKinna et al., 2021). The steps of our BA analysis are as follows:

– Calculate the paired mean (M) for all matched observations for the chosen cloud parameter,

$$M_i = \frac{X_i + Y_i}{2}, \tag{1}$$

where:

$M_i$ = Paired mean for $i$th observation,

$X_i$ = Ground $i$th observation,

$Y_i$ = Satellite $i$th observation.

– Calculate the normalized differences (D) for all matched observations for the chosen cloud parameter,

$$D_i = \frac{X_i - Y_i}{(\sigma_{ci}^2 + \sigma_{ai}^2)^{1/2}}, \tag{2}$$

where:

$D_i$ = Normalised difference for $i$th observation,



$\sigma_{ci}$ = Cloud_CCI pixel uncertainty for $i$th observation, in the ORAC optimal estimation,

$\sigma_{ai}$ = Standard deviation observed in the ground-based measurements for the temporal averaged period for $i$th observation.

– Plot M vs. D, and calculate the linear regression coefficients (Pearson coefficient, $r$) with the significance testing (t-test). The corresponding formula for $r$ is:

$$r = \frac{\sum_{i=1}^{n}(X_i - \bar{x})(Y_i - \bar{y})}{\sqrt{\sum_{i=1}^{n}(X_i - \bar{x})^2}\sqrt{\sum_{i=1}^{n}(Y_i - \bar{y})^2}}, \tag{3}$$

where:

$\bar{x}$ = Mean of ground observations,

$\bar{y}$ = Mean of satellite observations,

$n$ = the total number of observations.

– Check whether the Pearson coefficient from the above step is insignificant at 95% confidence level. BA bias (B) and Limits Of Agreements (LOA) are defined in this case. The BA bias is the mean normalized difference (D), and the LOA denotes the two-sigma region for the observed BA bias.

$$B = \sum_{i=1}^{n}\frac{D_i}{n}, \tag{4}$$

$$LOA = [B - 2 * B_s, B + 2 * B_s]. \tag{5}$$

where:

$B_s$ = Standard deviation in $D_i$,

– If the significance test fails, then B and LOA are considered meaningless and are not calculated. In this case, the bias is dependent on the paired mean, which implies that the bias changes significantly with the paired mean, i.e., a scale-dependent or variant case.

– We consider that an ideal distribution of normalized differences would have a bias of 0 and 95% of the values lie within $\pm 2 * (\sigma_{ci}^2 + \sigma_{ai}^2)^{1/2}$. The obtained results from the BA plots (B, LOA) are interpreted by comparing them to this assumed ideal range.

## 4 Results and Discussion

### 4.1 Cloud mask

The initial step in any cloud retrieval scheme is to apply a cloud mask to the satellite observations. This classifies the pixels to be cloud or cloud-free. If a cloud is identified in a pixel, cloud data products are retrieved. The Cloud_CCI, as discussed





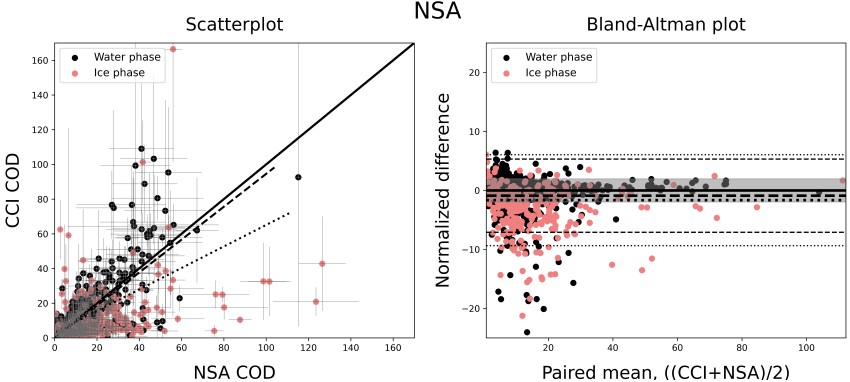

**Figure 3.** Left: Ground-based COD vs. Cloud_CCI COD daytime data products plotted for the period 2010 to 2018 at NSA. The data points are classified by liquid water (black) and ice (red) phases as given by the Cloud_CCI cloud product. The thick black line is the 1-1 line. The regression lines for the liquid water phase cases (dashed) and for all cases (liquid water + ice; dotted) are shown as well. The error bars represent the standard deviation in the 5-minute averaging period of the ground-based measurements and the Cloud_CCI pixel uncertainty. Right: Bland-Altman plot showing the paired mean vs. the normalized difference for the liquid water phase cases (black dots) and for ice cases (red dots). The thick dashed line indicates the BA bias for the liquid cases with the thin dashed lines indicating the corresponding Limits of Agreements (LOA). Correspondingly, the dotted lines indicate the results including all cases (liquid water + ice). The shaded area indicates the assumed ideal range. Table 1 lists the corresponding statistics.

| COD | All cases | Ice phase | Liquid water phase |
|---|---|---|---|
| Samples | 920 | 287 | 633 |
| Pearson correlation (p-value) | 0.53 (< 0.01) | 0.40 (<0.01) | 0.72 (< 0.01) |
| Linear regression slope (std. err.) | 0.62 (0.03) | 0.31 (0.04) | 0.95 (0.04) |
| Linear regression intercept (std. err.) | 3.09 (0.73) | 4.81 (1.21) | -0.24 (0.66) |
| Median bias (CCI-NSA) | -4.6 | -8.8 | -2.8 |
| LOA | [-9.3, 6.1] | [- , -] | [-7.1, 5.3] |
| BA bias | -1.6 | - | -0.8 |
| Percentage of |D| > 1.96 | 40% | 59% | 32% |

**Table 1.** Statistics related to Fig. 3, NSA vs Cloud_CCI COD data for all cases (liquid water + ice), ice phase, and liquid water phase for the period 2010-2018. The std. err. represents the standard error. The LOA and BA bias represents the Limits of Agreements and Bland-Altman bias respectively as calculated from Eqs. 4 & 5 respectively. The last row represents the percentage of values outside the assumed ideal range.



| LWP | Hyytiälä | NSA | Ny-Å | Summit |
|---|---|---|---|---|
| Samples | 58 | 282 | 166 | 211 |
| Pearson correlation (p-value) | 0.78 (< 0.01) | 0.66 (< 0.01) | 0.64 (< 0.01) | 0.63 (< 0.01) |
| Linear regression slope (std. err.) | 0.84 (0.09) | 0.97 (0.06) | 0.54 (0.05) | 0.30 (0.10) |
| Linear regression intercept (std. err.) | 21.10 (11.52) | 30.14 (6.74) | 57.13 (7.65) | 12.37 (4.86) |
| Median bias (CCI-Ground) $gm^{-2}$ | 7.1 | 15.6 | 23.9 | $\approx 5$ ($\leq 20gm^{-2}$), -13.4 ($> 20gm^{-2}$) |
| LOA | [-3.5, 3.7] | [-2.8, 4.3] | [-3.5, 4.3] | [-,-] |
| BA bias | 0.1 | 0.7 | 0.4 | - |
| Percentage of |D| > 1.96 | 11% | 29% | 30% | 9% |

**Table 2.** Statistics as in Table 1 but related to Fig.4 showing ground-based vs Cloud_CCI LWP data for all liquid cases according to the Cloud_CCI product.

before, uses an ANN trained with CALIOP data to identify the cloudy and cloud-free pixels. Fig. 2 shows the plot of SW CF vs. SW COD from the NSA, stratified by Cloud_CCI cloud mask information. The COD from NSA is divided into three ranges as follows, less than 3 OD units (optically thin), 3-10 OD units, and more than 10 OD units. The black and red color
shows the cloudy and clear conditions, respectively, as described in the Cloud_CCI data. The shaded region of Fig. 2 comprises ground-based measurements having CF values of 50-100%. We choose the 50-100% values of CF to investigate whether the Cloud_CCI cloud mask is accurate. For these data, we expect the Cloud_CCI cloud mask to be cloudy when the NSA CF is at least 0.5. The *hit* percentage for a particular COD range is defined as the number of Cloud_CCI cloudy pixels that occurs in the shaded region (NSA CF > 0.5) divided by the total number of pixels in the shaded region. The *miss* percentage is defined as
the number of Cloud_CCI clear pixels that occurs in the shaded region (NSA CF > 0.5) divided by the total number of pixels in the shaded region. In this study, we use the *hit* and *miss* percentages to quantify the success of the cloud mask. In Fig. 2, the hit percentage is $\approx 75\%$ for the optically thin clouds, whereas, for the optical thick clouds, it is as high as 90%. For the clouds having optical depths greater than 10 OD units, the hit percentage is greater than 95%. The miss-percentage is around 25% for the optically thin clouds, i.e., for COD $\leq 3$. The miss-percentage was reduced to almost 10% for the clouds, whose optical
depths are between 3 and 10 OD units. The relatively high miss-percentage for the thin clouds is attributed to the presence of unidentified cirrus clouds. These are challenging to detect in the retrievals using passive remote sensing satellite-borne sensors. Another possible explanation for the miss-percentages is the presence of fog or clouds that form as a result of a temperature inversion in the boundary layer. Such conditions are more frequent in the Arctic, compared to other regions (Tjernström et al., 2019; Wang et al., 2022). For optically thick clouds whose COD is more than 10 OD units, the miss-percentage is as small as
5%. In summary, the CCI cloud mask shows cloudy for 90% of the cases when the ground-based COD $\geq 3$ OD units.





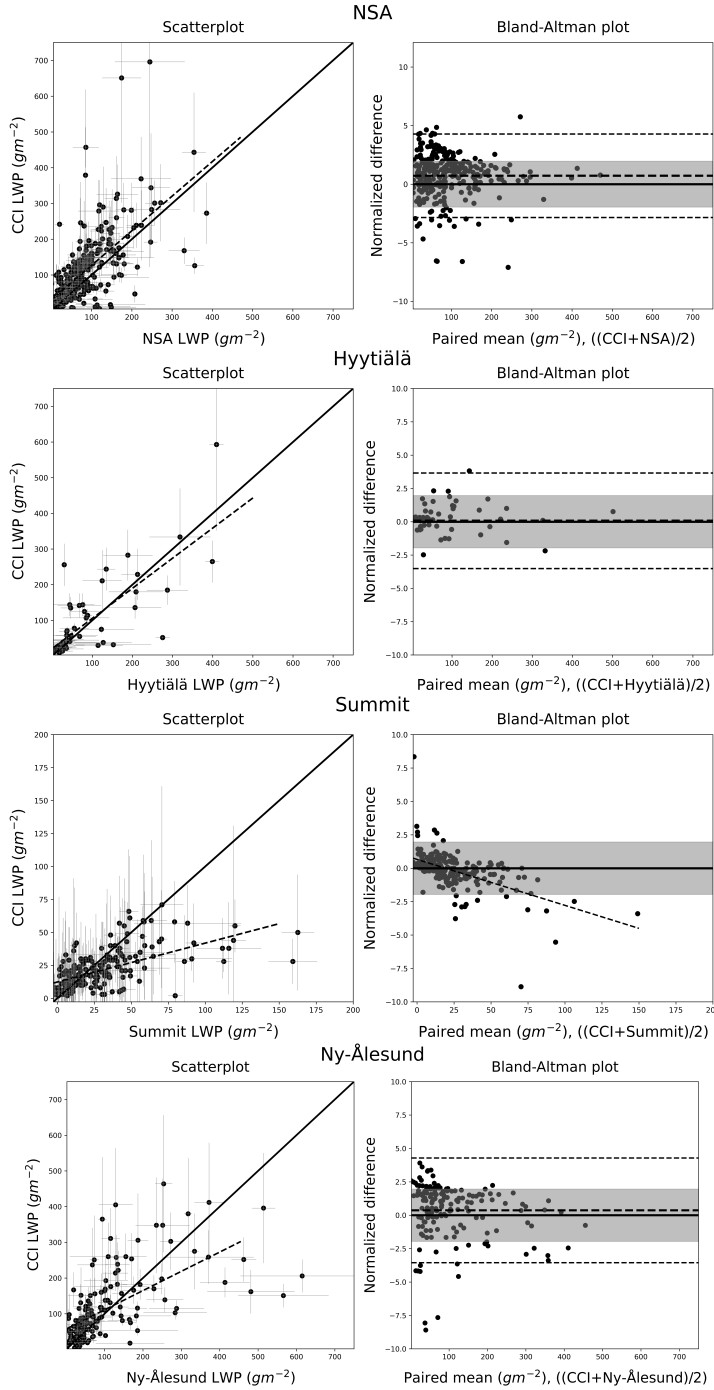

**Figure 4.** Similar to Fig.3 but for LWP over Hyytiälä (2014), NSA (2010-2018), Ny-Å (2016-2018), and Summit (2010-2018) for all liquid cases according to the Cloud_CCI product. Table 2 lists the corresponding statistics. Note that the dashed line in the B-A plot for Summit is the regression line.





## 4.2 Cloud optical thickness and liquid water path

The validation of Cloud_CCI COD is performed for daytime cases at NSA (ground-based COD is only available at NSA) because the retrieved COD from the instruments uses visible channels. Cloud_CCI COD is classified as either liquid water or ice, as explained in the Cloud_CCI data product. The statistical analysis (i.e. the determination of regression coefficients and

BA biases) of the comparison of COD is calculated for the three possible types of COD: all types (liquid water phase + ice phase), the ice phase, and the liquid water phase. Fig. 3 shows the regression lines and BA biases for all types and liquid water phase types because scale-independence is not observed for the ice phase type. From Fig. 3 and Table 1, it can be seen that the COD retrieved by Cloud_CCI for the liquid water phase clouds shows a better agreement compared to the NSA measurements than for the ice phase clouds. The regression coefficients are improved significantly when only liquid water phase clouds are

considered, i.e. the regression line for the liquid water phase clouds is closer to the 1-1 line. The Cloud_CCI COD product underestimates the COD: for water phase clouds by 2.8 OD units and by 8.8 OD for ice phase clouds. In general, liquid water clouds whose COD is less than 30 OD units are closer to the 1-1 line, and for ice phase clouds, the Cloud_CCI COD values are much lower than the NSA measurement values (red color points below the 1-1 line). In summary, the Cloud_CCI product underestimates COD by around 4.6 OD units. There is no scale dependence for COD observed for the liquid water phase

clouds. The BA bias for liquid water phase clouds is close to 0 (-0.8) and becomes almost twice that magnitude (-1.6) with the inclusion of ice phase clouds (i.e., the all case scenario). The LOA for the liquid water clouds is also closer to the ideal range (the shaded region) and becomes wider when augmented with the ice phase clouds. The percentage of cases outside the ideal bias range is also better for liquid water phase clouds than ice phase clouds. In the all-case type, 40% of the measurements lie outside the ideal range and most of these are due to the ice phase clouds below 30 OD units where an underestimation by

Cloud_CCI is clearly visible. This value is reduced to 32% when only liquid water phase clouds are considered.

LWP is further analyzed when the cloud phase is liquid according to the Cloud_CCI product. Figure 4 and Table 2 show that Cloud_CCI generally overestimates the LWP when compared with the ground-based radiometer measurements. The LWP comparison for Hyytiälä, NSA and Ny-Å reveals similar results: for Hyytiälä and NSA, LWP is overestimated by 7 gm$^{-2}$ and 15 gm$^{-2}$ respectively (bearing in mind the number of samples over Hyytiälä is quite small). At Ny-Å, LWP is overestimated by

about 24 gm$^{-2}$. The comparison statistics for the measurements made at Hyytiälä are generally better than those at other sites. The overestimation of LWP is most probably due to the uncertainty in the CER retrievals. The CER, especially in the Arctic, is not necessarily constant with respect to altitude within the cloud. The inhomogeneity of the CER cannot be accounted for in the retrieval. The LWP statistics for Summit are inferior to those at the other three sites. As expected (see, for instance, Bennartz et al. (2013)), very low values of LWP are observed over the Greenland ice shield. Unlike the other three sites,

a scale-dependence bias is observed approximately at about 20 gm$^{-2}$, which can be observed from the BA plot (where the regression line intersects the zero-line). Thus, the median biases are calculated for pixels when LWP is $\leq$ 20 gm$^{-2}$ and for LWP > 20 gm$^{-2}$. At LWP values $\leq$ 20 gm$^{-2}$, we observe an overestimation of $\approx$ 5 gm$^{-2}$ in the Cloud_CCI data whereas for LWP > 20 gm$^{-2}$, there is a negative bias i.e. underestimation of about 13 gm$^{-2}$, which, according to the BA plot, is strongly driven by the lower range of LWP values (20 gm$^{-2}$ $\leq$ LWP $\leq$ 50 gm$^{-2}$). Because of this scale-dependence, LOAs and BA biases are



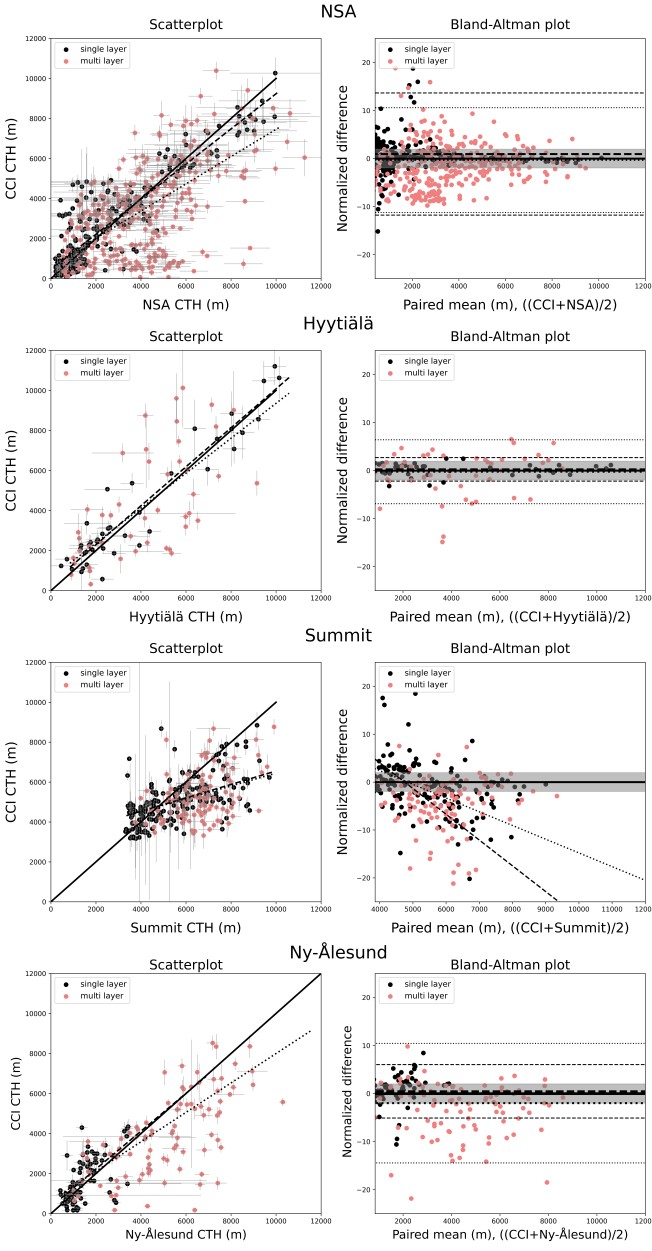

**Figure 5.** Similar to Fig.3 but for CTH over Hyytiälä (2014), NSA (2010-2018), Ny-Å (2016-2018), and Summit (2010-2018). Tables 3 and 4 lists the corresponding statistics. The data points are classified by single-layer (black) and multi-layer (red) clouds as identified from the ground-based observations. Left: The regression lines for the single-layer cloud cases (dashed) and for all cloud cases (single- and multi-layer; dotted) are shown as well. Right: The BA bias and Limits of Agreements (LOA) are shown for single-layer cloud cases (dashed) and all cloud cases (dotted). Note that the dashed (dotted) line in the BA plot for Summit represents the regression line for single-layer cloud (all cloud) cases instead.





| CTH | Hyytiälä | | NSA | |
|---|---|---|---|---|
| | All cases | Single layer | All cases | Single layer |
| Samples | 93 | 41 | 615 | 284 |
| Pearson correlation (p-value) | 0.81 (< 0.01) | 0.93 (< 0.01) | 0.74 (< 0.01) | 0.91 (< 0.01) |
| Linear regression slope (std. err.) | 0.87 (0.07) | 0.97 (0.05) | 0.68 (0.04) | 0.88 (0.02) |
| Linear regression intercept (std. err.) | 634.15 (314.36) | 346.22 (242.51) | 650.03 (99.49) | 458.83 (75.31) |
| Median bias (CCI-Ground) (m) | -180 | 240 | -208 | 197 |
| BA bias | -0.23 | 0.26 | -0.32 | 0.94 |
| LOA | [-3.73, 3.27] | [-1.02, 1.55] | [-6.06, 5.42] | [-5.74, 7.63] |
| Percentage of |D| > 1.96 | 39% | 12% | 49% | 32% |

**Table 3.** Statistics as in Table 1 but related to Fig. 5 showing ground-based vs Cloud_CCI CTH data over Hyytiälä (2014) and NSA (2010-2018).

| CTH | Ny-Å | | Summit | |
|---|---|---|---|---|
| | All cases | Single layer | All cases | Single layer |
| Samples | 178 | 85 | 327 | 197 |
| Pearson correlation (p-value) | 0.77 (< 0.01) | 0.62 (< 0.01) | 0.45 (< 0.01) | 0.52 (< 0.01) |
| Linear regression slope (std. err.) | 0.64 (0.04) | 0.90 (0.12) | 0.32 (0.03) | 0.34 (0.04) |
| Linear regression intercept(std. err.) | 693.61 (161.16) | 359.67 (202.83) | 3236.57 (217.17) | 3189.66 (231.17) |
| Median bias (CCI-Ground) (m) | -172 | 187 | 284 (<= 5000 m) -1371 (> 5000 m ) | 379 (<= 5000 m) -1099 (> 5000 m ) |
| BA bias | -2.06 | 0.84 | - | - |
| LOA | [-13.65, 9.52] | [-2.50, 4.18] | [- , -] | [- , -] |
| Percentage of |D| > 1.96 | 59% | 49% | 64% | 60% |

**Table 4.** Statistics as in Table 1 but related to Fig. 5 showing ground-based vs Cloud_CCI CTH data over Ny-Å (2016-2018) and Summit (2010-2018).



not calculated for Summit. Scale-independence was observed over Hyytiälä, NSA, and Ny-Å leading to the consideration of
       BA bias and LOAs. These LOAs are much closer to the defined ideal range where the BA bias is less than 1. The percentage
       of points outside the ideal range is also small for all the sites i.e. 11% for Hyytiälä, 29% for the NSA, 30% for the Ny-Å,
       and 9% for Summit. In previous studies (Sporre et al., 2016), the LWP determined from passive satellite-based remote sensing
       depends on the SZA and larger biases are observed at higher SZA. For the Cloud_CCI LWP data product, there is no sign of

change in statistics, when separating the days based on SZA (not shown). The biases observed at all the sites are smaller than
       the uncertainty of the ground-based LWP retrievals.

## 4.3   Cloud top height

The comparison is separated into single-layer and multi-layer conditions using the lidar-radar classification. A single-layer
cloud is defined when the ground-based instruments showed only one cloud layer throughout the averaged 5-minute interval.

Similarly, if two or more cloud layers are observed, the scene is defined as a multi-layer cloud. For Ny-Å, multi-layer clouds are
       assumed if at least 2 vertical bins (i.e. 40 m) are cloud-free between the cloud layers while for the other sites one bin (i.e. 45 m)
       is at least cloud-free between the cloud layers. Fig. 5 shows the scatter and BA plots for the CTH data. The statistics (Table 3 &
       4) are calculated separately for all cloud types, defined as the sum of single and multi-layer clouds, and single-layer clouds. The
       regression line for all cloud types is much closer to the one-one line for Hyytiälä (slope = 0.87) than those for NSA (0.68), Ny-

Å (0.64), and Summit (0.32). However, it should be kept in mind that the sample size for Hyytiälä is small. Overall the selected
       sites, the CTH values for the single-layer clouds are in better agreement with the ground-based measurements than those for the
       multi-layer clouds. The single-layer CTH is overestimated at all sites. When single-layer clouds are combined with multi-layer
       clouds, i.e. all cloud types, the Cloud_CCI CTH underestimates the ground-based CTH implying an underestimation of CTH of
       multi-layer clouds. This overestimation for single-layer clouds and underestimation for multi-layer clouds is similar for all sites

except for Summit where the underestimation of CTH for multi-layer clouds is not so large. In brief: the overestimation for the
       single-layer CTHs at Hyytiälä, NSA, and Ny-Å are ≈ 240 m, ≈ 197 m, and ≈ 187 m respectively, while CTHs of all clouds are
       underestimated by ≈ 180 m, ≈ 208 m, and ≈ 172 m at Hyytiälä, NSA, and Ny-Å respectively (Table 3 & 4). This difference
       between the overestimation of single-layer CTH and underestimation of all cloud CTH over Hyytiälä, NSA, and Ny-Å is
       around 420 m, 405 m, and 360 m respectively. This suggests that the presence of multi-layer clouds adds an underestimation

by about 360-420 m in the Cloud_CCI CTH retrievals. We attribute this underestimation of the CTH of multi-layer clouds to
       the reduced sensitivity of the passive space-borne instruments to detect high-altitude thin clouds, which are often comprised of
       ice. A thick cloud deck underneath thin high-altitude clouds will contribute significantly to the observed radiance at top of the
       atmosphere. Similar to the analyses of the LWP, the analysis of the CTH data from the Summit station (located at ca. 3.2 km
       above m.s.l) showed different statistics and high retrieval uncertainties for the Cloud_CCI CTH as compared to the values at

other stations (Table 4). Over Summit, scale-dependent bias is observed with an overestimation of CTH values below 5000 m
       and an underestimation above 5000 m (derived from the intersection of the regression line with the zero-line in the BA plot).
       This led us to report the median biases for these two separate cases. For clouds whose CTH is below 5000 m over the Summit,
       the overestimation of CTH for single-layer clouds is around 380 m. In contrast to the other sites, no distinct underestimation of





the multi-layer CTH is observed over Summit. For CTH values above 5000 m, CTH is largely underestimated by Cloud_CCI, i.e. by 1100 m for single-layer clouds and by 1371 m for all cloud types (single and multi-layer clouds).

From the BA plots over Hyytiälä, NSA, and Ny-Å, no observable scale-dependent bias for the CTH retrievals is noticed, leading to a meaningful LOA. The LOA over Hyytiälä is much closer to the defined acceptable range and within the defined acceptable range for single-layer CTHs. Compared to Hyytiälä, over NSA and Ny-Å, the LOA is wider for both considered types, i.e., all cloud CTHs and single-layer CTHs, indicating more significant differences between satellite and ground-based
CTH values. These differences are clearly identifiable in the BA plot, where 32% of such cases fall outside the defined acceptable range for single-layer clouds (Table 3). The number of cases falling outside the defined acceptable range for single-layer clouds is 12% over Hyytiälä. In contrast, for multi-layer clouds, the samples outside the defined acceptable range are about 40%, 50%, and 60% over Hyytiälä, NSA, and Ny-Å, respectively. For Summit, a high amount of values ($> 60\%$) are outside the defined acceptable range, resulting from many CTH values that are underestimated. The BA bias and LOA are not calculated
for Summit since they showed scale-dependence.

Previous studies from Holz et al. (2008) and Sporre et al. (2016) for MODIS and VIIRS satellite retrievals observed similar patterns of overestimation of CTH for the single-layered clouds and underestimation for multilayered clouds. These studies did not include high Arctic sites such as NSA and Summit. The plausible reason reported in those studies is the differences in the cloud optical thicknesses between different cloud layers, which then influences the accuracy of the multi-layer CTH retrievals.

**5  Conclusions**

The Climate Change Initiative (CCI) global cloud products (Cloud_CCI) are one of the publicly available climate data records. Cloud_CCI makes use of the measurements from the AVHRR PM dataset, which extends over more than three decades. To our knowledge, this is the first study to validate Cloud_CCI data products by comparison with measurements from ground-based instruments in the Arctic ($> 60°$ N), and one of few to validate satellite cloud products over the Arctic.
The comparisons of the Cloud_CCI and ground-based cloud data during the sunlit months, available for Hyytiälä, NSA, Ny-Å, and Summit, in most cases, showed high Pearson correlation coefficients, PCC, $\geq 0.6$, having a 99% confidence level. The Cloud_CCI cloud mask indicates cloudy for more than 90% of the time in the presence of optically thick clouds (COD $> 10$) having CF $> 0.5$ as indicated from ground-based observations. 10% of clouds are missed by Cloud_CCI for cases with COD in the range of 3 to 10. We conclude that the CCI cloud mask is reasonably well validated for optically thick clouds
over NSA. Such clouds are found in the oceanic regions of the Arctic, such as the Barents Sea, Laptev Sea, Kara Sea, East Siberian Sea, and the Chukchi Sea, where optically dense clouds are present most of the time during the sunlit months (Kay et al., 2016; Huang et al., 2021). As a result of the warming Arctic, we expect cloud properties to change over time. We, therefore, need sufficiently accurate satellite cloud data records to detect these changes. To this end, we need to understand whether the differences between the satellite data and ground-based data are scale independent or dependent. Consequently,
the Bland-Altman approach has been used in this study to investigate this issue. If the differences in the cloud properties between the satellite and ground-based data do not significantly increase/decrease with the value of the cloud property itself,





they are assumed to be scale-independent. The majority of the Cloud_CCI results have no scale biases when compared with the ground-based instruments, except for Summit. In general, the Cloud_CCI underestimates COD by 3 OD units for liquid water phase clouds. These results imply that the Cloud_CCI COD values when compared with the ground-based radiometers

are expected to have differences $\leq 3$ OD units over NSA and similar regions where liquid water phase cloud formation is more active during sunlit months (Gu et al., 2021). The overestimation of LWP at high latitude sites, such as NSA and Ny-Å, is two to three times higher than at the low latitude Arctic site Hyytiälä. At NSA, Ny-Å and Hyytiälä, the Cloud_CCI CTH retrievals overestimate the single-layer cloud CTH but underestimate the CTH of multi-layer clouds. The comparison over Summit, Greenland, showed a different CTH behavior compared to the other three stations. At Summit, significant scale-dependent

biases are observed in single-layer clouds and for all cloud types (single + multi-layer). These issues of high CTH biases over Greenland and the effect of multi-layer clouds need to be taken into account when using the Cloud_CCI CTH data products to evaluate models and for other applications. In general, the AVHRR PM Cloud_CCI cloud data for LWP are within the ground-based instrument uncertainties, and Cloud_CCI CTH agrees within 500 m compared to the ground-based lidar-radar observations. There are some other exceptions, in particular for Summit, as discussed in the results section. In particular, the

Cloud_CCI data products at high altitudes in Greenland and multi-layer clouds are associated with larger errors.

We conclude that further studies are needed comparing ground-based and satellite-borne measurements, not only to extend this study in terms of temporal and spatial coverage but also to investigate further the applicability of passive space-based remote sensing instruments for cloud retrievals in the Arctic.

*Author contributions.*   KV and MV conceptualized the study. KV did the analysis for the validation and wrote the manuscript with contribu-

tions from the co-authors. LL and MV suggested the Bland-Altman method. MS contributed to the Cloud_CCI data. MDS contributed to the Summit data record. KE contributed to the Ny-Å data record. JPB provided guidance for the work. All authors were involved in examining and reviewing the results. All authors were involved in editing the paper.

*Competing interests.*   The authors have no competing interests to declare.

*Acknowledgements.*   We gratefully acknowledge the funding by the Deutsche Forschungsgemeinschaft (DFG, German Research Foundation)

– project no. 268020496 – TRR 172, within the Transregional Collaborative Research Center "ArctiC Amplification: Climate Relevant Atmospheric and SurfaCe Processes, and Feedback Mechanisms (AC)[3]". The study has also in part been funded by the University of Bremen. LL was partly funded by the Alexander von Humboldt foundation via the Feodor-Lynen fellowship 2020. The contribution of MS was supported by the European Space Agency (ESA) through the Cloud cci project (contract no.: 4000128637/20/I-NB). Data from NSA and Hyytiälä were obtained from the Atmospheric Radiation Measurement (ARM) User Facility, a U.S. Department of Energy (DOE) Office

of Science User Facility Managed by the Biological and Environmental Research Program. Data from Summit were obtained from the





ICECAPS project funded by the U. S. National Science Foundation (OPP-1801764, OPP-1801318, OPP-1801477). We also acknowledge Kirk Knobelspiesse (NASA GSFC) for the provision of statistical routines.

*Code and data availability.* CCI Cloud data are available at https://public.satproj.klima.dwd.de/data/ESA_Cloud_CCI/CLD_PRODUCTS/ v3.0/L3U/AVHRR-PM/. Data from NSA and Hyytiälä were obtained from the Atmospheric Radiation Measurement (ARM) User Facility, a

U.S. Department of Energy (DOE) Office of Science User Facility Managed by the Biological and Environmental Research Program. Data from Summit were obtained from the ICECAPS project. Ny-Å measurements are taken from the AWIPEV atmospheric observatory. The validation code are based on python and available on request. The Bland-Altman analysis is done using K. Knobelspiesse's github project https://github.com/knobelsp/BlandAltman/





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
