# Peer review of "Validation of the Cloud\_CCI cloud products in the Arctic"

_Atmospheric Measurement Techniques, 2022_

## Author Comment (AC2)

**Response to the reviewer**

[Figure]

Fig 1) COD validation over NSA in log scale and normal scale

[Figure]

Fig 2) Temporal variations of LWP over NSA, Summit and Ny-Ålesund (NyA)

[Figure]

Fig 3) Temporal variations of CTH over NSA, Summit and Ny-Ålesund (NyA)

[Figure]

[Figure]

Fig 4) Temporal variations of COD (all cases) and COD (only liquid clouds) over NSA

---

## Author Response (AR1)

The following sections correspond to the point-by-point response to the reviews followed by changes made in manuscript

**1) The remarks by the reviewers are in red, and our answers are in black.**

RC1

Q1) Line 60: The term "reasonable agreement" does not contain much useful information. Please quantify this "reasonable agreement".

A1) We have now modified and quantified the statement as follows. (Line 58-62)

*"In addition, some studies, such as Sporre et al. (2016), have compared LWP and CTH retrievals from Moderate Resolution Imaging Spectroradiometer (MODIS) and Visible Infrared Imaging Radiometer Suite (VIIRS) with the measurements of the Atmospheric Radiation Measurement (ARM) mobile facility at the high-latitude site in Hyytiälä. The results from this study showed an LWP difference of < 15 gm-2 and CTH differences of < 500 m between satellite and ground-based observations. However, the CTH differences are more than 1000 m for the clouds located above 6000 m."*

Q2). Line 62: Please give a number for the high viewing angles for which cloud detection problems happen.

A2). The line is now changed as follows with the observed threshold high viewing angles. (Lines 65)

*"Earlier versions of the MODIS measurements were compared with ARM measurements over the North Slope of Alaska (NSA), where the authors identified cloud detection problems at high viewing angles (> 55$^0$)."*

Q3) Lines 118-130: Here the COD product, among others, is described. Although mentioned later, please state that the COD refers to the SW (visible) COD.

A3) Changed accordingly

Q4) Line 157: Change ``from an MWR" to ``from MWR".

A4) Changed accordingly

Q5) Lines 171-188: Somewhere in this paragraph mention whether the CTH refers to the height above sea level or height above ground level.

A5) Thank you for the suggestion, added the following line to the paragraph. (Lines 188-189)

*"The CTH value, which corresponds to the height above ground level, is converted to the height above mean sea level to match satellite CTH values."*

Q6) Lines 272-274: It is claimed that Fig. 3 ``... shows better agreement ...". Most of the points in the scatter plot is lumped together for COD smaller than about 40. This collection of points

hardly shows better agreement as claimed. To better show the behavior of the points for COD<40 maybe try using log scale or add a zoom-in-plot for this region.

A6) We tested the logarithmic plot, but it creates very large error bars for low values (less than 5 OD units), attracting a lot of unnecessary attention (see below, left) We propose to use a zoom-in plot, as shown below.

[Figure]

Fig 1) COD validation over NSA in log scale and normal scale

Q7) Line 276: Please state that these underestimates are biases.

A7) Thank you for the suggestion, changed it accordingly

Q8) Lines 288-289: For the COD all decimal values from Table 1 were included when reporting biases (line 276) For the LWP the decimal values are omitted compared to Table 2. Is there a reason for this? If not please be consistent. In any case, 15.6 for NSA rounds up to 16 and not down to 15.

A8) Thank you for pointing to this inconsistency. We have now rounded all biases to integers.

Yes, 15.6 is now rounded to 16. Thank you for noticing

Q9) Lines 315-317: Please clarify this sentence.

A9) Lines 315-317 in the previous version include some extra information mentioning the fewer sample points observed at Hyytiälä (since the data record is one year) as compared to the other. Now, in the current version, we changed the sentence and made it much clearer as follows.

*"The CTH values for the single-layer clouds are in better agreement with the ground-based measurements than those for the multi-layer clouds. In general, the single-layer CTH is overestimated at all sites."*

Q10) Lines 351-383: Based on your findings: do you find the Cloud_CCI cloud products to be of such a quality to be useful in studies of the Arctic Amplification? If not, why not? And how can it be improved to be useful?

A10) Yes, with limitations.

The Arctic is a challenging environment for satellite retrievals of clouds. In this study, we presented the biases in satellite retrievals and determined how they differ quantitatively from ground-based cloud products. From the results, it is clear that the CCI dataset is not the best choice in regions where multilayered clouds occur frequently. Another important limitation is the presence of mixed-phase clouds over the Arctic, which are currently not distinguished in CCI cloud phase products.

On the other hand, the CTH generally showed good results, especially for the low-level clouds that are common in the Arctic. The LWP biases are within the uncertainties of the ground-based instruments, which makes them a useful parameter for understanding phenomena such as Arctic amplification, and in this way, the dataset has already provided important insights for cloud studies in the Arctic (Lelli et al. (2023)).

One way to improve on the above could be to use the CALIOP polarization data in the initial steps of CCI generation to distinguish cloud phases. Furthermore, distinguishing single-layer and multilayer clouds is an important improvement opportunity

General comment: This study assumes that there are no trends in cloud products due to instrument drifts or other changes. May you please comment on this? Also, have the temporal behavior in the differences between the ground-based and Cloud_CCI products been looked at? This may be done by for example plotting the differences between ground-based and Cloud_CCI cloud products as a function of time.

A) Earlier, Trishchenko et al. (2002), Wang and Cao (2008) emphasized that suboptimal radiometric calibration of the AVHRR thermal channels could lead to inconsistencies, which then lead to discrepancies in the detection of clouds in the Arctic, the surface radiative fluxes and their trends (Zygmuntowska et al., 2012). Before creating the satellite data set, each PM sensor was cross-calibrated with well-functioning sensors. The scanning imaging Absorption SpectroMeter for Atmospheric CHartographY (SCIAMACHY) served as the spectral reference for the visible wavelengths and the Infrared Atmospheric Sounding Interferometer (IASI) for the thermal channels (Stengel et al., 2020). This led to improved cloud parameters (Sus et al., 2018; McGarragh et al., 2018) in terms of precision, accuracy and stability (Stengel et al., 2017). In a recent Arctic cloud study by Lelli et al. (2023), the CCI dataset has been used and no bias-induced errors in the trend derivation were found. Further, the Bland-Altman concept used in this study shall help to understand whether the biases change significantly with the true value (in short: will the biases observed for COD 10-20 differ from those for COD 21-50?). We would have expected, that, if there were such remaining instrumental biases, the Bland-Altman approach would have revealed them,

Regarding the second point: The plots where ground-based and Cloud_CCI cloud products are plotted as a function of time for the sites where long-term data is available are shown from Fig 2-4.

RC2

Q1) Not including time component seems to be a missed opportunity. Why not show a few evaluation metrics as a function of time, esp for those stations where the longer measurements are available? Given that the meteorological and thermodynamical conditions (and thus their impact on cloud properties) do vary significantly among the sunlit months in the Arctic, it would be really useful to the users to understand the performance of ESA-CCI-Cloud products during the various sunlit months.

Q2) Another aspect related to the point above would be to express bias or metrics as a function of solar zenith angle and/or viewing zenith angle.

The following answers correspond to both Q1 and Q2

A1) As part of the preparation of the manuscript, we also looked at time series. Although it is a 9-year period (2010-19) over NSA, a large number of samples are not suitable for direct comparison with ground-based instruments because the cost functions of the satellite observations are too high. This is a fairly expected scenario for most passive satellite sensors over the Arctic. However, the main problem is that the days that are eligible for validation are not the same over the entire period. This condition has prevented us from directly comparing the time series or trend assessment. E.g., in 2010 we obtained the samples for the first days of April when the melt starts and the clouds are visually dense, but in the following year, 2011, the samples may have been selected somewhat after this regular Arctic activity, which may not representative as in 2010.

However, below are the plots, Fig. 1-3, in which temporal comparisons were made between the satellite and ground-based instruments for the sites for which long-term data are available. From the results, it can be seen that the slopes of the annual variations have the same sign for both satellite and ground-based instruments. The increase in LWP from 2015-2017 over NSA is very well captured. In most cases (> 95%), the CCI median values are within 50% of the ground-based distributions. A CTH decline of about 100 m/yr observed in the ARM NSA is consistent with the CCI. The observed slope values of ground-based COD (and COD equivalent), which are ~0.6 units/year, are also consistent with satellite observations (note that these are not statistically significant though).

[Figure]

Fig 2: Temporal variations of LWP over NSA, Summit and Ny-Ålesund (NyA)

[Figure]

Fig 3: Temporal variations of CTH over NSA, Summit and Ny-Ålesund (NyA)

[Figure]

Fig 4: Temporal variations of COD (all cases) and COD (only liquid clouds) over NSA

A2) SZA/VZA: The separation based on SZA does not provide thresholds that justify good/bad results. One of the main reasons could be using a "best pixel" with the lowest SZA in the L3U grid products.

Q3) Near-isothermal conditions in the lower boundary layer and temperature inversions make it notoriously difficult to place clouds at the right height in the passive retrievals. Is this also the case for CCI-Clouds retrievals? If so, there is not much discussion or investigation of this aspect in the CTH evaluation.

A3) Undoubtedly, the mentioned aspects lead to the complication of the situation. Exactly these aspects have led us to this validation study. We wanted to know how good the quality of the cloud products of the dataset is under Arctic conditions, although it is actually a global retrieval. In terms of cloud top, we were surprised by how well the results agreed. From our point of view, the situation for the CTH over Summit is not directly related to the general Arctic conditions, but rather to the special ones over Greenland: optical, relatively thin clouds which have a rather low cloud top (Bennartz et al., 2013) and is potentially affected by temperature inversions at the very surface (~2m) and higher above (Adolph et al., 2018).

Q4) What role does the AVHRR detection sensitivity actually play when you stratify the results in Fig. 2 according to COD? The results presented in Fig. 12 in Karlsson and Håkansson (2018) are relevant here and should be discussed.

A4) Thanks for the valuable suggestion, it is indeed relevant and needs to be discussed. From Fig. 12, 13 in Karlsson and Håkansson (2018), the minimum optical thickness required for cloud detection ranges from ~ 0.5 to ~ 4.5 depending on the region. In our case, which is over Alaska, it almost meets the threshold values. A new line is added in the current version 261-263.

Q5) Are there more data available from the ground-based measurements taken in the ACTRIS framework in the Arctic? If so, they would also be useful here.

A5) Thank you for the recommendation, ACTRIS is useful. But the cloud parameters that are relevant to us, cover similar stations that we have targeted.

Q6) Were there mixed-phase clouds detected in the ground-based retrievals? If so, how are those samples handled?

A6) This study does not address mixed-phase clouds in either satellite or ground-based measurements. We are aware that this limits our study to some degree. However, this limitation applies to most cloud research in the Arctic and is a general problem due to the lack of coverage of measurement data.

**2) Changes made in the manuscript**

- We would like to point out an error that has been corrected in the manuscript: The CTH comparison period for NSA and Summit are from 2012-2016 and 2010-2014, respectively.
- While discussing in the text, all decimals are rounded, and in the tables, values are consistent to two decimals wherever they're required.
- A study referring to the validation of TROPOMI cloud products is added in the *introduction* (Compernolle et al., 2021). **Lines 55-57**
- **Lines 61-63, Lines 63-65** are more quantified (as per RC1 Q1, Q2 suggestions)
- **Lines 188-189** are added explaining the height used is the height above the mean sea level.
- Fig 3 in the manuscript has been changed with a zoom-in plot for COD values less than 40, and an explanation for it is given in the caption.
- **Line 215** is rewritten to make it more explanatory.
- In *results*, 4.1, cloud mask section, a study by Karlsson and Håkansson, 2018, is referred (as redirected by RC2, Q4). **Lines 261-263**
- ' $\approx$ ' symbol is used wherever necessary.
- In the *Acknowledgements* section co-author MDS Mercator Fellowship as part of (AC)3 is added. **Line 399**
- In the *Code and data availability* section, lines regarding summit data are changed. **Line 405**
- In the *References* missing DOIs are added.

**References:**

Adolph, A. C., Albert, M. R., and Hall, D. K.: Near-surface temperature inversion during summer at Summit, Greenland, and its relation to MODIS-derived surface temperatures, The Cryosphere, 12, 907–920, https://doi.org/10.5194/tc-12-907-2018, 2018.

Karlsson, K.-G. and Håkansson, N.: Characterization of AVHRR global cloud detection sensitivity based on CALIPSO-CALIOP cloud optical thickness information: demonstration of results based on the CM SAF CLARA-A2 climate data record, Atmos. Meas. Tech., 11, 633–649, https://doi.org/10.5194/amt-11-633-2018, 2018.

Lelli, L., Vountas, M., Khosravi, N., and Burrows, J. P.: Satellite remote sensing of regional and seasonal Arctic cooling showing a multi-decadal trend towards brighter and more liquid clouds, Atmospheric Chemistry and Physics, 23, 2579–2611, https://doi.org/10.5194/acp-2023-2579-2023, 2023.

Linke, O., Quaas, J., Baumer, F., Becker, S., Chylik, J., Dahlke, S., Ehrlich, A., Handorf, D., Jacobi, C., Kalesse-Los, H., Lelli, L., Mehrdad, S., Neggers, R. A. J., Riebold, J., Saavedra Garfias, P., Schnierstein, N., Shupe, M. D., Smith, C., Spreen, G., Verneuil, B., Vinjamuri, K. S., Vountas, M., and Wendisch, M.: Constraints on simulated past Arctic amplification and lapse-rate feedback from observations, Atmos. Chem. Phys. Discuss. [preprint], https://doi.org/10.5194/acp-2022-836, in review, 2023.

McGarragh, G. R., Poulsen, C. A., Thomas, G. E., Povey, A. C., Sus, O., Stapelberg, S., Schlundt, C., Proud, S., Christensen, M. W., Stengel, M., et al.: The Community Cloud retrieval for CLimate (CC4CL)–Part 2: The optimal estimation approach, Atmospheric Measurement Techniques, 11, 3397–3431, 2018.

Stengel, M., Stapelberg, S., Sus, O., Schlundt, C., Poulsen, C., Thomas, G., Christensen, M., Carbajal Henken, C., Preusker, R., Fischer, J., et al.: Cloud property datasets retrieved from AVHRR, MODIS, AATSR and MERIS in the framework of the Cloud_cci project, Earth System Science Data, 9, 881–904, 2017.

Stengel, M., Stapelberg, S., Sus, O., Finkensieper, S., Würzler, B., Philipp, D., Hollmann, R., Poulsen, C., Christensen, M., and McGarragh, G.: Cloud_cci Advanced Very High Resolution Radiometer post meridiem (AVHRR-PM) dataset version 3: 35-year climatology of global cloud and radiation properties, Earth System Science Data, 12, 41–60, 2020.

Sus, O., Stengel, M., Stapelberg, S., McGarragh, G., Poulsen, C., Povey, A. C., Schlundt, C., Thomas, G., Christensen, M., Proud, S., et al.: The Community Cloud retrieval for CLimate (CC4CL)–Part 1: A framework applied to multiple satellite imaging sensors, Atmospheric Measurement Techniques, 11, 3373–3396, 2018.

Trishchenko, A. P., Fedosejevs, G., Li, Z., and Cihlar, J.: Trends and uncertainties in thermal calibration of AVHRR radiometers onboard NOAA-9 to NOAA-16, Journal of Geophysical Research: Atmospheres, 107, ACL–17, 2002.

Wang, L. and Cao, C.: On-orbit calibration assessment of AVHRR longwave channels on MetOp-A using IASI, IEEE Transactions on Geoscience and Remote Sensing, 46, 4005–4013, 2008.25